# Weighted Conformal Prediction for Time-Dependent PDEs

## Abstract

Uncertainty quantification is crucial in scientific machine learning, where models inform safety-critical tasks such as flood forecasting, aerodynamic optimization, and financial risk management. Conformal prediction provides distribution-free coverage guarantees, but in time-dependent settings common to physics and engineering, these guarantees can break down, leading to systematic undercoverage. We study this problem in the context of surrogate models for time-dependent physical systems described by partial differential equations (PDEs). We prove that in a function space setting, distributions at arbitrarily close times can be mutually singular, making exact coverage guarantees impossible. As a solution, we facilitate weighted conformal prediction for a broad class of PDE problems arising from discretized models and validate these results in experiments. While prior work often sidesteps time dependence—by assuming exchangeability, focusing on short horizons, or ignoring long-term deployment—we address it directly by providing exact coverage guarantees through reweighting calibration scores.

## 1 Introduction

Many problems in physics and engineering, such as weather prediction, aerodynamics, and financial modeling, are governed by partial differential equations (PDEs). Classical numerical solvers are accurate but computationally expensive, scaling poorly with dimensionality or repeated simulations. AI-based surrogate models have emerged as a promising alternative, providing fast approximations of PDE solutions. Prominent examples include physics-informed neural networks (PINNs) (Raissi et al., 2019), DeepONets (Lu et al., 2021), and neural operators (Anandkumar et al., 2019; Li et al., 2021). Most notably, neural operators have demonstrated remarkable success in generalizing across different discretizations, geometries, and boundary conditions.

Despite these advances, surrogate models still lack principled mechanisms for uncertainty quantification. This limitation is critical, since scientific and engineering decisions often depend on reliable confidence assessments of model outputs. Conformal prediction (CP) Vovk et al. (2022) provides a principled framework, producing distribution-free uncertainty sets with guaranteed marginal coverage. These guarantees, however, rely on exchangeability between calibration and test samples—a condition that is frequently violated in time-dependent PDEs.

**Non-Stationarity in Time-Dependent PDEs.** Let $u_t$ denote the solution of a time-dependent PDE at time point $t$. In practice, we are interested in predicting $u_{t+\delta}$ for several time steps $\delta > 0$, beyond the available training and calibration data. Unless $u_t$ is a stationary process, test samples follow a different distribution than observed calibration samples, breaking the exchangeability assumption required by conformal prediction.

This type of non-stationarity is ubiquitous: sudden shocks (e.g., stock market crashes), long-term structural changes (e.g., climate trends), and limited development windows (e.g., laboratory testing) all produce systematic shifts in the data distribution (see figure 1). Even for simple PDEs, the marginal distribution of $u_t$ may drift continuously in $t$ and diverge arbitrarily as $t \to \infty$.

**Implications for Conformal Prediction.** The consequence is that conformal intervals calibrated at time $t$ may undercover at future times $t + \delta$. Figure 2 illustrates this behavior on the backward heat equation. In the top row, calibration at time step $\delta$ still yields valid coverage at $3\delta$. In contrast,

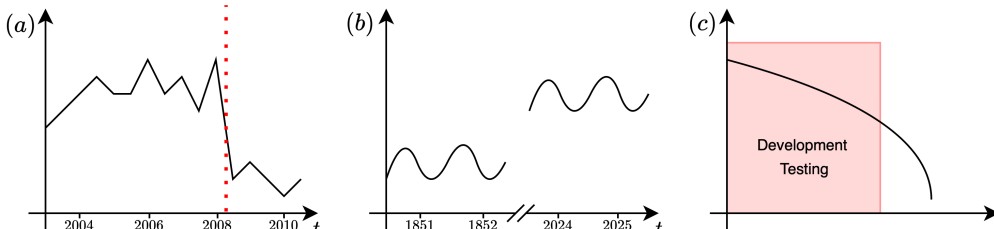

Figure 1: Examples of different types of non-stationarity that arise in time-dependent systems. *(a)* Sudden distribution shift due to external shocks, illustrated by a stock market crash. *(b)* Long-term structural changes, as in climate time series. *(c)* Limited observation window during system development, where calibration and testing occur on different parts of the trajectory.

calibration at step $4\delta$ (bottom row) leads to undercoverage already at the first prediction time step, because the PDE grows progressively unstable. This demonstrates that simply tuning the discretization step does not resolve the problem: coverage degradation is inherent to the temporal drift.

A concrete example arises in weather forecasting, where calibration on short-term simulations may produce intervals that appear reliable but fail to capture rare extreme events at later horizons. In such cases, nominal $90\%$ coverage can collapse well below the target, producing forecasts that systematically underestimate risk.

Consequently, CP coverage guarantees do not hold in time-dependent PDEs. While there are first methods to sidestep the non-exchangeability (see section 2), all of these come with limiting assumptions that prohibit broad applicability.

**Our Contributions.** In this work, we address this gap by studying CP for time-dependent surrogate models of PDEs, providing the following contributions:

1. We analyze the function-space formulation of the learning problem and show that even in simple settings, such as the heat equation, the total variation (TV) distance is maximal for any time distance. This shows that a pure function-space perspective, as often used in the neural operator literature, is unsuitable for the non-exchangeable CP framework.

2. For a broad class of PDEs, we derive explicit densities for the discretized solutions over time, facilitating the use of *weighted conformal prediction*. This enables exact coverage guarantees for PDEs without limiting assumptions on their time-dependent behavior.

3. We empirically validate our method on several time-dependent PDEs and compare it to alternative CP approaches (which assume exchangeability or local exchangeability). We show that these limiting assumptions on the time dynamics indeed lead to undercoverage, and that our approach is the only method providing reliable coverage over time.

The paper is structured as follows. In section 2, we review related work. Section 3 provides background on CP, PDEs, and surrogate models. In section 4, we formalize the problem setting, present our result on function spaces, and our weighted CP framework. Section 5 presents empirical results demonstrating the effectiveness of our approach and section 6 concludes.

## 2 RELATED WORK

**Trajectory-Based Exchangeability.** The most straightforward option to bypass the exchangeability issue is to treat entire trajectories as the exchangeable units. Moya et al. (2025) use DeepONets to predict full solution trajectories, calibrating CP on trajectory-level samples. This avoids assumptions on exchangeability within the calibrated time horizon, but does not address potential distribution shifts beyond this horizon, e.g. in a potential model deployment. Gray et al. (2025) follow the same strategy, though their method applies to arbitrary surrogate models beyond neural operators. Gopakumar et al. (2025) also adopt trajectory-level calibration, but focus on conformal sets for deviations between surrogates and the governing PDE operator, rather than for the solution itself.

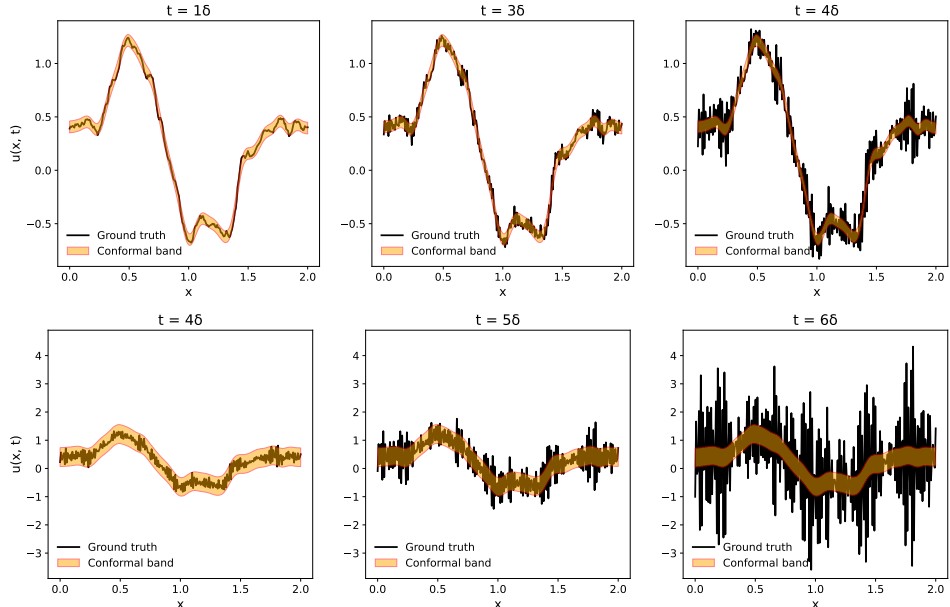

Figure 2: CP bands when calibrating at different time resolutions of the backward heat equation. Each figure shows the solution $u(x, t)$ and CP bands over the spatial domain $x$ at one time point. *(Top)* Calibration on data from time step $\delta$: prediction bands remain valid even at step $3\delta$. *(Bottom)* Calibration on data from time step $4\delta$: undercoverage occurs already after one time step.

**Relaxed Exchangeability Assumptions.** Motivated by data scarcity, Gopakumar et al. (2024) go beyond trajectory-based exchangeability and construct calibration samples by slicing long time series into shorter segments. This construction implicitly assumes that the time series is stationary across segments, which may hold approximately in periodic systems (e.g., weather data), but fails in general dynamical systems with non-periodic trends or sustained drifts.

Harris & Liu (2025) take a different approach with their Local Spectral Conformal Inference (LSCI) method, designed for neural operators. Instead of global exchangeability, they derive conformal bands with guarantees under *local exchangeability*, i.e., that points close in time are nearly exchangeable. When local exchangeability holds, LSCI provides the first principled way for time-adaptive prediction sets with coverage guarantees for neural operators. However, validating this assumption in practice is usually not feasible (see appendix A.1). Therefore, Harris & Liu (2025) *assume* local exchangeability in their experiments by taking very small time steps. In Figure 2, however, we see one example where a calibration on time step 1 leads to good empirical coverage for three further time steps, but when calibrating again at a later time, coverage already drops after one time step because the solution gets exponentially noisier. Thus, tuning the step size at calibration does not ensure local exchangeability at test time.

**Time Series Conformal Prediction** Time-series conformal prediction is an active research area. Most existing methods provide only asymptotic guarantees (e.g., Gibbs & Candès (2021); Angelopoulos et al. (2023); Xu & Xie (2023)), meaning they achieve coverage on average over infinite time steps or only in the limit as the time horizon grows. Another line of work seeks subclasses of time series with structural properties that enable per–time-step coverage. For example, Oliveira et al. (2022) shows coverage for stationary, $\beta$-mixing processes. Our approach falls into this latter category, and to the best of our knowledge, no prior work leverages PDE-specific properties to obtain conformal coverage.

## 3 BACKGROUND

### 3.1 CONFORMAL PREDICTION

Conformal prediction (CP) is a framework for constructing prediction sets with marginal finite-sample coverage guarantees Vovk et al. (2022). In the standard split setting, a model is trained on $\mathcal{D}_{\text{train}}$ and calibrated on $\mathcal{D}_{\text{cal}}$, yielding a set-valued predictor $\mathcal{C}$ such that, for a test sample $(\mathrm{x}, \mathrm{y})$,

$$\mathcal{P}\big(\mathrm{y} \in \mathcal{C}(\mathrm{x})\big) \ \geq \ 1 - \alpha$$

at coverage level $1 - \alpha$. This guarantee relies on **exchangeability** of calibration and test samples—that is, their joint distribution is invariant under permutations. When exchangeability is violated, coverage may fail.

**Conformal Prediction Beyond Exchangeability.** When calibration and test distributions differ but are related by a likelihood ratio, weighted CP provides a natural extension Vovk et al. (2022); Barber et al. (2023). In this setting, calibration samples are reweighted by

$$w_i \ \propto \ \frac{p_{\text{test}}(x_i)}{p_{\text{cal}}(x_i)}, \qquad \sum_i w_i = 1,$$

so that the conformal quantile is computed with respect to these weights. Here, the index $i$ ranges over all calibration data points and the target test point. If the density ratio is known or can be estimated, weighted CP can restore exact coverage in covariate-shift settings. In our PDE setup, the linear–Gaussian structure allows us to compute these ratios in closed form, enabling precise conformal bands (see section 4.4).

In case a closed-form evaluation of the weights is not possible, Barber et al. (2023) provide corrections for the conformal guarantees based on the TV distance[1] between calibration and test distribution, that hold even in the general case of non-exchangeability:

$$\mathcal{P}\big(\mathrm{y} \in \mathcal{C}(\mathrm{x})\big) \ \geq \ 1 - \alpha - \sum_{i=1}^{n} w_i \, d_{\text{TV}}(\mathbf{z}, \mathbf{z}^i),$$

where $\mathbf{z} = ((\mathbf{x}_1, \mathbf{y}_1), \ldots, (\mathbf{x}_{n+1}, \mathbf{y}_{n+1}))$ for calibration samples $((\mathbf{x}_1, \mathbf{y}_1), \ldots, (\mathbf{x}_n, \mathbf{y}_n))$ and test point $(\mathbf{x}_{n+1}, \mathbf{y}_{n+1})$ and $\mathbf{z}^i$ arises from permuting the test point with the $i^{\text{th}}$ calibration point.

For further details, we recommend the summary by Angelopoulos et al. (2024).

### 3.2 PDEs AS OPERATOR MAPPINGS

Many dynamical systems in physics and engineering can be described by evolution equations of the form

$$\frac{\partial u}{\partial t}(\boldsymbol{x}, t) = \mathcal{L}_{\boldsymbol{x}} u(\boldsymbol{x}, t),$$

where $u : \Omega \times [0, \infty) \to \mathbb{R}$ is the state variable, $\boldsymbol{x} \in \Omega \subset \mathbb{R}^d$ denotes spatial coordinates, $t \geq 0$ is the time, and $\mathcal{L}_{\boldsymbol{x}}$ is a (possibly nonlinear) differential operator acting on the spatial variable $\boldsymbol{x}$. We write $u_t := u(\cdot, t)$ for the spatial slice at time $t$. In this paper, we are interested in the Cauchy-type problem, where we consider boundary conditions on $\bar{\Omega}$ and initial conditions $u_0(\boldsymbol{x})$ from some Banach space of functions $(\mathcal{A}, \| \cdot \|_{\mathcal{A}})$ and are interested in a solution $u_t(\boldsymbol{x})$ in some Banach space of functions $(\mathcal{U}_t, \| \cdot \|_{\mathcal{U}_t})$. Typically, $u_t : \Omega \to \mathbb{R}$ and $u_t \in L^2(\Omega)$. We will further only consider well-posed problems, where we can define solution operators

$$\mathcal{G}_t : \mathcal{A} \to \mathcal{U}_t, \ \mathcal{G}_t(\boldsymbol{a})(\boldsymbol{x}) \mapsto u(\boldsymbol{x}, t)$$

that uniquely map an initial condition to a solution function $u_t(\boldsymbol{x})$ and the map $t \mapsto \mathcal{G}_t$ is continuous in $t$. In the rest of the paper, we will assume that all functions come from the same space, so $\mathcal{A} = \mathcal{U}_t$ for all $t \geq 0$, to simplify the notation, but the results apply more generally.

---

[1]The TV distance is originally defined on probability measures, and whenever we write $d_{\text{TV}}(\mathrm{x}, \mathrm{y})$ for random variables x and y, or $d_{\text{TV}}(\mathcal{P}_{\mathrm{x}}, \mathcal{P}_{\mathrm{y}})$ for probability distributions $\mathcal{P}_{\mathrm{x}}$ and $\mathcal{P}_{\mathrm{y}}$, we refer to the TV distance between their corresponding probability measures.

### 3.3 SURROGATE MODELS

**Physics-Informed Neural Networks (PINNs).**   PINNs (Raissi et al., 2019) approximate PDE solutions by training a neural network to satisfy both observed data and the underlying PDE. The loss function penalizes violations of the differential operator $\mathcal{L}$ and boundary/initial conditions, so that the neural network implicitly encodes the solution $u(\boldsymbol{x}, t)$. PINNs are flexible and require only point-wise evaluations of the PDE residual, but they often struggle with stiff dynamics, sharp gradients, or long time horizons.

**Deep Operator Networks (DeepONets).**   DeepONets (Lu et al., 2021) aim to directly learn non-linear operators between function spaces. They decompose the problem into a *branch net*, which encodes the input function (e.g., the initial condition), and a *trunk net*, which encodes the query point $(\boldsymbol{x}, t)$. The outputs are combined to approximate $u(\boldsymbol{x}, t) = \mathcal{G}_t(\boldsymbol{a})(\boldsymbol{x}, t)$. DeepONets provide a general framework for operator learning and can handle diverse geometries and boundary conditions, but require large and representative training data.

**Neural Operators.**   Neural operators Anandkumar et al. (2019); Li et al. (2021) generalize this idea further by parameterizing mappings $\mathcal{G}$ directly in function space, rather than through point-wise regression. Unlike standard neural networks, which approximate finite-dimensional mappings, neural operators approximate $\mathcal{G}$ itself and can generalize across discretizations. In practice, functions are observed on a finite set of points (grids or meshes), and the learned operator is evaluated on these (or other) discretizations. Popular variants include the Fourier Neural Operator, which uses spectral convolutions for global context, and the Graph Neural Operator, which extends to irregular meshes.

**Other Surrogates.**   Beyond these, there are also kernel-based approaches, reduced-order models, and Gaussian process surrogates. However, in the machine learning literature, PINNs, DeepONets, and neural operators have emerged as the three most prominent classes of PDE surrogates.

## 4 WEIGHTED CONFORMAL PREDICTION FOR TIME-DEPENDENT PDE SURROGATE MODELS

### 4.1 PROBLEM SETTING FOR CONFORMAL PREDICTION ON TIME-DEPENDENT PDES

To apply CP in the PDE setting, we start by specifying the underlying structure.

**From Initial Conditions to Solutions.**   Assume we have an analytical form of the PDE, so that we can generate our own data using numerical solvers. We first focus on the case where we want to predict the solution at one fixed time point $t$ for a given initial condition. To obtain our training data $\mathcal{D}_{\text{train}}$, we would sample initial conditions $u_{0,i} \sim \mathcal{P}_0$, $i = 1, \ldots, N_{\text{train}}$, from a distribution on $\mathcal{U}$, and obtain the corresponding solution at time $t$ by numerically solving the PDE. This defines a pushforward measure[2]

$$\mathcal{P}_t := (S_t)_{\#}\mathcal{P}_0,$$

where $S_t : \mathcal{U} \to \mathcal{U}$ is the PDE solution operator mapping initial conditions $u_0$ to solutions $u_t$. Our training dataset then consists of

$$\mathcal{D}_{\text{train}} = \{(u_{0,i}, u_{t,i})\}_{i=1}^{N_{\text{train}}}, \quad u_{t,i} \sim \mathcal{P}_t.$$

If we now consider consecutive time points, our distribution changes over time:

$$\mathcal{P}_0 \xrightarrow{S_\delta} \mathcal{P}_\delta \xrightarrow{S_\delta} \mathcal{P}_{2\delta} \xrightarrow{S_\delta} \cdots$$

Thus, we obtain a sequence of probability distributions $\{\mathcal{P}_t\}_{t \geq 0}$ on the same function space, evolving under the PDE dynamics.

**Implication for Conformal Prediction.**   Calibration and test data drawn from different $\mathcal{P}_t$ are therefore **not exchangeable**: although they live in the same function space, their distributions shift with time.

---

[2]We slightly abuse notation here by writing the pushforward in terms of the distribution instead of the measure corresponding to the distribution.

## 4.2 DISTRIBUTION SHIFTS IN FUNCTION SPACES

Having specified the problem setup, we now investigate if we can calculate the TV distance between the laws of a PDE solution at different time points. If the TV distance of the laws of time points $t$ and $t + \delta$ were moderate, we could recover CP coverage guarantees for the $t + \delta$ prediction using the approach from Barber et al. (2023).

We will start by analyzing the problem in the function-space setting, as is often employed in the neural operator literature and related CP works (Harris & Liu, 2025; Gray et al., 2025; Mollaali et al., 2024). We will show that even for a simple PDE, like the heat equation with Gaussian initial distribution, the TV distance between the solution-distributions $\mathcal{P}_t, \mathcal{P}_{t+\delta}$ at two time points $t, t + \delta$ is always maximal,

$$d_{\mathrm{TV}}(\mathcal{P}_t, \mathcal{P}_{t+\delta}) = 1, \quad \text{for all } t \geq 0, \ \delta > 0.$$

This is representative of a broader phenomenon that "[...] measures in infinite-dimensional spaces have a strong tendency of being mutually singular." Hairer (2023). As a direct consequence, regular CP—and any method relying on equality or even approximate similarity between calibration and test distributions—becomes inapplicable. Regaining guarantees would require stronger implicit biases, but this lies beyond the scope of this paper.

Finally, note that, while this issue complicates theoretical considerations in the neural operator literature, it is not necessarily problematic for practical CP on surrogate models. In practice, we always work with finite-dimensional discretizations, which mitigate this effect, as will be discussed in section 4.3.

**Theorem 4.1.** *Consider the one-dimensional heat equation on the domain $\Omega = (0, 1)$ with Dirichlet boundary conditions*

$$\begin{aligned}
\frac{\partial u}{\partial t}(x, t) &= \frac{\partial^2 u}{\partial x^2}(x, t), & x \in (0, 1), \ t \geq 0, \\
u(0, t) &= u(1, t) = 0, & t \geq 0, \\
u(x, 0) &= u_0(x), & x \in (0, 1),
\end{aligned}$$

*where $u : \bar{\Omega} \times [0, \infty) \to \mathbb{R}$ denotes the temperature at location $x$ and time $t$. Suppose the initial condition is sampled from a Gaussian distribution*

$$\mathcal{P}_0 \sim \mathcal{N}(\mathbf{0}, (\boldsymbol{I} - \boldsymbol{\Lambda})^{-1}),$$

*where $\boldsymbol{\Lambda}$ is the Laplace operator on $\Omega$ with Dirichlet boundary conditions. Then, for any $t \geq 0$, $\delta > 0$, the TV distance between the measures $\mathcal{P}_t$ and $\mathcal{P}_{t+\delta}$ of the solution $u(\cdot, t)$ and $u(\cdot, t + \delta)$ is maximal, i.e.*

$$d_{\mathrm{TV}}(\mathcal{P}_t, \mathcal{P}_{t+\delta}) = 1.$$

The proof is provided in appendix A.2.

We will now discuss how, despite the issue above, coverage guarantees can be recovered for time-dependent PDE surrogate models in practice.

## 4.3 RECOVERING COVERAGE GUARANTEES

The following theorem provides the exact distribution of the solution $u_t$ on a discretized space, using the *method of lines*. We provide an intuitive example in appendix A.3.

**Theorem 4.2.** *Let $\Omega \subset \mathbb{R}^d$ be a bounded domain, and let*

$$\mathcal{M} := \{x_1, \ldots, x_n\} \subset \Omega$$

*denote a discretization of $\Omega$. Consider the finite-difference scheme in space, with $\boldsymbol{A} \in \mathbb{R}^{n \times n}$ approximating the solution of*

$$\frac{\partial u}{\partial t}(\boldsymbol{x}, t) = \mathcal{L}_{\boldsymbol{x}} u(\boldsymbol{x}, t), \qquad \boldsymbol{x} \in \Omega, \ t \geq 0,$$

*with linear boundary conditions on $\partial \Omega$, where $\mathcal{L}_{\boldsymbol{x}}$ is a linear spatial differential operator. This yields the discretized dynamics*

$$\frac{d\boldsymbol{u}(t)}{dt} = \boldsymbol{A}\boldsymbol{u}(t) + \boldsymbol{r}(t), \quad \boldsymbol{u}(t), \boldsymbol{r}(t) \in \mathbb{R}^n.$$

*Suppose the initial condition satisfies $\boldsymbol{u}(0) \sim \mathcal{N}(\boldsymbol{\mu}_0, \boldsymbol{\Sigma}_0)$. Then, for $t \geq 0$ and $\delta > 0$, the law $\mathcal{P}_t$ of $\boldsymbol{u}(t)$ is Gaussian with mean*

$$\boldsymbol{\mu}_t = \exp(t\boldsymbol{A})\boldsymbol{\mu}_0 + \int_0^t \exp((t-s)\boldsymbol{A})\boldsymbol{r}(s)\mathrm{d}s$$

*and covariance*

$$\boldsymbol{\Sigma}_t = \exp(t\boldsymbol{A})\boldsymbol{\Sigma}_0 \exp(t\boldsymbol{A}^T).$$

*Proof.* As we discretized only in space, not in time, the finite difference scheme yields a linear system of ODEs

$$\frac{d\boldsymbol{u}(t)}{dt} = \boldsymbol{A}\boldsymbol{u}(t) + \boldsymbol{r}(t).$$

As $\boldsymbol{A}$ is independent of $t$ and $\boldsymbol{r}(t)$ is the deterministic source term, the solution of the system of ODEs is given by

$$\boldsymbol{u}(t) = \exp(t\boldsymbol{A})\,\boldsymbol{u}(0) + \int_0^t \exp((t-s)\boldsymbol{A})\boldsymbol{r}(s)\mathrm{d}s.$$

Note that we assumed $\boldsymbol{u}(0)$ is Gaussian, i.e.,

$$\boldsymbol{u}_0 \sim \mathcal{N}(\boldsymbol{\mu}_0, \boldsymbol{\Sigma}_0), \quad \boldsymbol{\mu}_0 \in \mathbb{R}^n, \boldsymbol{\Sigma}_0 \in \mathbb{R}^{n \times n},$$

and $\exp(t\boldsymbol{A})$ is just a matrix, so $\boldsymbol{u}(t)$ is also Gaussian with mean $\boldsymbol{\mu}_t = \exp(t\boldsymbol{A})\boldsymbol{\mu}_0 + \int_0^t \exp((t-s)\boldsymbol{A})\boldsymbol{r}(s)\mathrm{d}s$ and covariance $\boldsymbol{\Sigma}_t = \exp(t\boldsymbol{A})\boldsymbol{\Sigma}_0 \exp(t\boldsymbol{A}^T)$.

$\square$

*Remark* 4.3. This result can be generalized to other initial distributions. The location-scale family of distributions, for example, is closed under affine transformations leading to similar results. The location-scale family includes, among others, the Gaussian, Cauchy, Laplace, and logistic distributions. Note, however, that the Gaussian assumption we made is the most common in recent literature (Li et al., 2021; Santos et al., 2023; Gopakumar et al., 2024; Zhou & Barati Farimani, 2025; Gopakumar et al., 2025). Also, from a physical viewpoint, a Gaussian random field aligns well with the laws of nature in the sense that the aggregate effect of many small independent perturbations, forming the initial condition, is approximately Gaussian by the central limit theorem. We added additional experiments, sampling from different location-scale initial distributions in the appendix A.8.

*Remark* 4.4. Theorem 4.2 also allows us to derive an upper bound on the TV distance of the laws of $u_t$ and $u_{t+\delta}$. While we will not make use of this result in our method, we provide the theorem and proof in appendix A.4.

### 4.4 LIKELIHOOD–WEIGHTED CONFORMAL PREDICTION

Theorem 4.2 shows that under a discretized linear PDE with Gaussian initial conditions, the solution at time $t$ is Gaussian with mean $\boldsymbol{\mu}_t$ and covariance $\boldsymbol{\Sigma}_t$ as stated in the theorem. Consequently, both calibration and test distributions (corresponding to time points $t$ and $t + \delta$ for one or more $\delta > 0$) are Gaussian and their density ratio is available in closed form. This enables a likelihood-weighted conformal predictor:

$$w_{i,\delta} \;\propto\; \frac{\mathcal{N}(\boldsymbol{u}_i;\; \boldsymbol{\mu}_{t+\delta}, \boldsymbol{\Sigma}_{t+\delta})}{\mathcal{N}(\boldsymbol{u}_i;\; \boldsymbol{\mu}_t, \boldsymbol{\Sigma}_t)}, \tag{1}$$

for all $\boldsymbol{u}_i$ belonging to the calibration set together with the target test point. Normalizing these weights and applying split CP with the weighted quantile yields conformal bands with formal coverage guarantees.

*Remark* 4.5. Within this CP framework applied to the discretized setting, we provide asymptotic—and in some cases even non-asymptotic—guarantees for the PDE solution $u(x,t)$ in the original space. The nature of the bounds depends on both the PDE and the discretization scheme, but the key idea is that the bands on the discretized solution can be transferred to the original solution by leveraging numerical error guarantees of the scheme.

## 5 EXPERIMENTS

**[NEW FIGURE]**

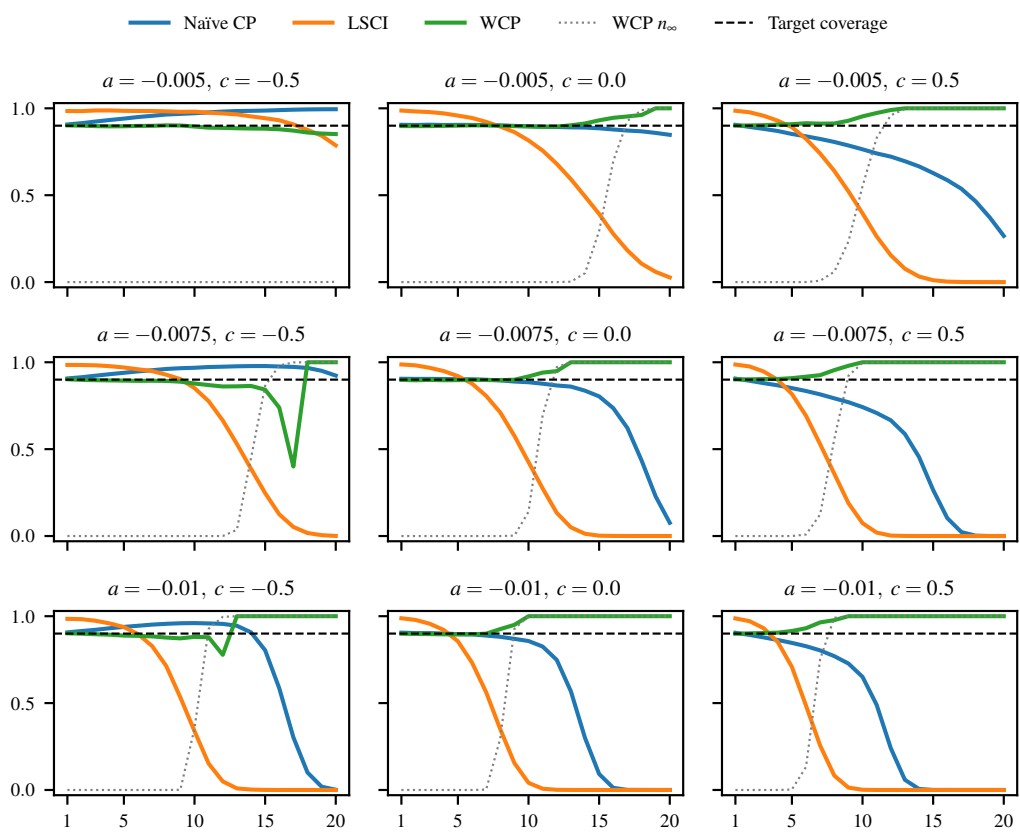

Figure 3: Mean coverages for varying $a, c$ and fixed $b = -0.5$ across increasing prediction horizon. We omit coverages when infinite conformal bands were reported (coverage of 1 would hold trivially). The 90% coverage target and, for WCP, the fraction of samples with infinite bandwidth $n_\infty$ are marked as black dashed line and gray dotted line, respectively.

**Experimental Setup**  We demonstrate our method on both synthetic and real-world data. For the synthetic case, we choose the general second order PDE framework

$$u_t(x,t) + au_{xx}(x,t) + bu_x(x,t) + cu(x,t) = 0, \qquad x \in (0,1),\ t \geq 0,$$
$$u(0,t) = u(1,t) = 0, \qquad t \geq 0,$$
$$u(x,0) = u_0(\boldsymbol{x}), \quad x \in (0,1).$$

where $a, b, c \in \mathbb{R}$ are tuneable parameters. To test WCP, we target setups where the PDE becomes more unstable over time ($a < 0$)—otherwise, CP methods that rely on observed residuals trivially cover at future time steps. Hence, we consider $a = \{-0.005, -0.0075, -0.01\}$ and for the remaining parameters we choose $b, c \in \{-0.5, 0, 0.5\}$. As a base model, we train a geometry-informed neural operator (Li et al., 2023) and calibrate on the residuals with the respective CP method (note

Table 1: Mean coverages and bandwidths over 5000 sampled initial conditions for varying $a$ and fixed $b = -0.5$, $c = -0.5$. For WCP, we also report the fraction of samples where infinite bands were reported ($n_\infty$) to maintain coverage guarantees. The gray font is chosen for better readability.

| | | | Timestep | | | | |
|---|---|---|---|---|---|---|---|
| | | | 1 | 5 | 10 | 15 | 20 |
| $a = -0.005$ | Naïve CP | Coverage | 0.91 | 0.94 | 0.97 | 0.99 | 0.99 |
| | | Bandwidth | 0.03 | 0.03 | 0.03 | 0.03 | 0.03 |
| | LSCI | Coverage | 0.98 | 0.99 | 0.98 | 0.94 | 0.79 |
| | | Bandwidth | 0.02 | 0.02 | 0.02 | 0.02 | 0.02 |
| | WCP (Ours) | Coverage | 0.9 | 0.9 | 0.9 | 0.88 | 0.85 |
| | | Bandwidth | 0.03 | 0.03 | 0.03 | 0.02 | 0.02 |
| | | $n_\infty$ | 0.0% | 0.0% | 0.0% | 0.0% | 0.2% |
| $a = -0.0075$ | Naïve CP | Coverage | 0.91 | 0.94 | 0.97 | 0.98 | 0.92 |
| | | Bandwidth | 0.03 | 0.03 | 0.03 | 0.03 | 0.03 |
| | LSCI | Coverage | 0.98 | 0.97 | 0.85 | 0.25 | 0.0 |
| | | Bandwidth | 0.02 | 0.02 | 0.02 | 0.02 | 0.02 |
| | WCP (Ours) | Coverage | 0.9 | 0.89 | 0.88 | 0.84 | 1.0 |
| | | Bandwidth | 0.03 | 0.03 | 0.03 | 0.03 | $\infty$ |
| | | $n_\infty$ | 0.0% | 0.0% | 0.0% | 86.4% | 100% |
| $a = -0.01$ | Naïve CP | Coverage | 0.91 | 0.94 | 0.96 | 0.8 | 0.0 |
| | | Bandwidth | 0.03 | 0.03 | 0.03 | 0.03 | 0.03 |
| | LSCI | Coverage | 0.98 | 0.94 | 0.34 | 0.0 | 0.0 |
| | | Bandwidth | 0.02 | 0.02 | 0.02 | 0.02 | 0.02 |
| | WCP (Ours) | Coverage | 0.9 | 0.89 | 0.88 | 1.0 | 1.0 |
| | | Bandwidth | 0.03 | 0.03 | 0.03 | $\infty$ | $\infty$ |
| | | $n_\infty$ | 0.0% | 0.0% | 35.4% | 100% | 100% |

that the choice of surrogate model is not important for downstream analysis). The task of the base model is to predict the solution $u_t$ at 20 time steps in the future. The task of the CP methods is to report conformal bands with 90% coverage. For each PDE, we sample 5000 trajectories to train the base model, 500 for validation, and 5000 for calibration and testing each. We adjusted the time steps and other parameters individually with more details in appendix A.5.

**Baselines** We define two baselines for our experiments. The first is a naïve implementation with no consideration of exchangeability (naïve CP). Specifically, we implemented Diquigiovanni et al. (2022), who define the score as the maximum absolute error over space and use the regular split CP algorithm. Since exchangeability does not hold in this setup, the conformal bands of naïve CP have **no formal guarantees**.

Secondly, we use the LSCI method (Harris & Liu, 2025) ($\lambda = 3$, projection dimension: 20, number of CP band samples: 5000). We choose a large number of band samples to push LSCI to over-coverage, so undercoverage can be evaluated in a fair manner. Note that because their guarantees only hold under the local exchangeability assumption which is not verifiable (see appendix A.1), the LSCI CP bands also have **no formal guarantees** in our experiments.

Our weighted conformal prediction (WCP) method is based on a weighted version of Diquigiovanni et al. (2022). Specifically, knowing that our solution is Gaussian at every time point, we weigh our score according to equation (1).

**Evaluation** For each method and each PDE, we report the mean coverage and bandwidth of the 5000 test set samples. We consider a sample covered if all of points of the function are within the conformal bands. In cases where the distributional dissimilarity of $u_t$ and $u_{t+\delta}$ is too large, our WCP method predicts infinite bands. If this is the case, we exclude the sample and only predict coverage of the other samples. We report the fraction of excluded samples $n_\infty$ in our results.

Note that reporting trivial bands is usually a more valuable result than delivering bands with undercoverage, especially in safety-critical tasks. The key strength of CP is its coverage guarantees and our WCP detects when it cannot predict meaningful bands and refrains from violating the target coverage.

**Results**   We report results for varying $a, c$ with $b = -0.5$ in Figure 3, and provide the corresponding plots for the remaining $b$-values in appendix A.7. For $b = c = -0.5$, the numerical results are listed in Table 1, while results for the other $c$-values are given in appendix A.7, together with a visualization of CP bands. Overall, in most configurations, naïve CP and LSCI fail to meet the coverage target—earlier and more severely as the PDE becomes noisier (i.e., for smaller $a$)—while WCP consistently meets its coverage guarantees. When $n_\infty$ approaches roughly $90\%$, WCP shows a slight drop in empirical coverage. This behavior is expected, as we only report coverage of non-trivial bands: with very few samples remaining, the empirical coverage is subject to higher stochastic noise. In practice, this can be addressed by using the bands only for sufficiently large remaining sample size or by considering the overall coverage including the trivial bands.

As discussed above, our method reports infinite bands for increasing distribution shift. Although this sacrifices meaningful bands, it ensures fully reliable coverage guarantees. Lastly, we observed that WCP and naïve CP are significantly faster than LSCI: When running LSCI on a MacBook Pro M4 Pro with 24GB RAM, sampling the conformal bands for 5000 test samples takes approximately 40 minutes. The WCP and the naïve method take only seconds. Overall, WCP is the only method providing **formal guarantees**, and we can see empirically that this is a clear advantage as soon as our system exhibits significant dynamics.

**Real-World Example**   To demonstrate the applicability of our method in real-world scenarios, we use the dataset of Wei et al. (2023). They provide a small 2D-dataset of pulsed-thermography measurements, where objects are heated and then cooled while surface temperatures are recorded to detect subsurface defects. We use only the cooldown phase, as it approximately follows the heat equation. We provide more details on our implementations and the results in appendix A.6. Our method achieves target coverage over all tested time steps.

## 6 DISCUSSION

Conformal prediction for time-dependent physical phenomena is often constrained by non-exchangeable data. In this work, we investigated whether coverage guarantees can be maintained beyond the exchangeability assumption. Our results show that this depends strongly on the setup. On function spaces, measures are typically mutually singular, making coverage guarantees unattainable. On discretized domains, however, we derived how weighted CP can be applied to linear PDEs to obtain coverage guarantees. We empirically validated that weighted CP is the only method that reliably achieves the target coverage compared to baselines.

These findings connect back to our starting point: non-stationarity in time-dependent PDEs breaks classical CP, but weighted CP offers a principled alternative. We established coverage for the class of linear PDEs. Although this class covers many practical problems, extending the analysis to nonlinear PDEs is a natural next step and would further broaden the applicability of conformal prediction in scientific machine learning.

REPRODUCIBILITY STATEMENT

We provided the code for the data generation, model training, fitting of conformal bands, and instructions on how to run it as supplementary material to the reviewers. With that, all figures and results can be reproduced independently. For the final version, we will set up a public GitHub repository. The proof for theorem 4.2 can be found in the main text, and the proof for theorem 4.1 can be found in appendix A.2.

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

## A  APPENDIX

### A.1  VALIDATING LOCAL EXCHANGEABILITY

To have formal guarantees with the LSCI approach from Harris & Liu (2025), it is required that the model's *residuals* $r_t$ are locally exchangeable. Considering this for the most simple setup of two time points $t$ and $t + \delta$ for some $\delta > 0$, this means that it must hold that

$$d_{\text{TV}}\big( \begin{pmatrix} r_t \\ r_{t+\delta} \end{pmatrix}, \begin{pmatrix} r_{t+\delta} \\ r_t \end{pmatrix} \big) \leq d(t, t + \delta)$$

for a pre-metric $d$ on the time domain. Note however, that since we do not have access to the laws of the random vectors above, we cannot reason about their TV distance. Even though in theorem 4.2 we derive the laws of the solutions $u_t$, it is not clear how to reason about the above left hand side without further assumptions (like independence of the residuals over time—which is not plausible as the solution has a clear time-dependence in all non-stationary PDEs).

## A.2 Proof of Theorem 4.1

**Theorem A.1.** *Consider the one-dimensional heat equation on the domain $\Omega = (0, 1)$ with Dirichlet boundary conditions*

$$\frac{\partial u}{\partial t}(x, t) = \frac{\partial^2 u}{\partial x^2}(x, t), \quad x \in (0, 1), \ t \geq 0,$$
$$u(0, t) = u(1, t) = 0, \quad t \geq 0,$$
$$u(x, 0) = u_0(x), \quad x \in (0, 1),$$

*where $u : \bar{\Omega} \times [0, \infty) \to \mathbb{R}$ denotes the temperature at location $x$ and time $t$. Suppose we sample the initial condition from a Gaussian distribution*

$$\mathcal{P}_0 \sim \mathcal{N}(\mathbf{0}, (\mathbf{I} - \mathbf{\Lambda})^{-1}),$$

*where $\mathbf{\Lambda}$ is the Laplace operator on $\Omega$ with Dirichlet boundary conditions. Then, for any $t \geq 0$, $\delta > 0$, the TV distance between the measures $\mathcal{P}_t$ and $\mathcal{P}_{t+\delta}$ of the solution $u(\cdot, t)$ and $u(\cdot, t + \delta)$ is maximal, i.e.*

$$d_{\mathrm{TV}}(\mathcal{P}_t, \mathcal{P}_{t+\delta}) = 1.$$

*Proof.* Notice that our whole setup is on a Hilbert Space. We begin by showing that the $(\mathbf{I} - \mathbf{\Lambda})^{-1}$ is a well-defined covariance operator on $L^2(\Omega)$. For this, according to Hairer (2023)[Proposition 4.17], the operator must be positive, symmetric and trace class. As both $\mathbf{I}$ and $-\mathbf{\Lambda}$ are positive and symmetric, so is their sum. As $-\mathbf{\Lambda}$ is positive, it follows that $(\mathbf{I} - \mathbf{\Lambda})$ is invertible, due to strictly positive eigenvalues. Further, the eigenvalues of $-\mathbf{\Lambda}$ with Dirichlet boundary conditions are given by $\lambda_n = (n\pi)^2$, with corresponding eigenfunctions $e_n(x) = \sqrt{2}\sin(n\pi x)$, which form an orthonormal basis of $L^2(\Omega)$. Therefore, the eigenvalues of $(\mathbf{I} - \mathbf{\Lambda})^{-1}$ are given by $\mu_n = 1/(1 + (n\pi)^2)$ and as

$$\sum_{n=1}^{\infty} \mu_n < \infty,$$

we conclude that $(\mathbf{I} - \mathbf{\Lambda})^{-1}$ is a trace class operator and thus defines a Gaussian measure on $L^2(\Omega)$.

Now our proof will be based on the Feldman-Hájek theorem Da Prato & Zabczyk (1992)[Theorem 2.23], which gives a characterization of when two Gaussian measures on a Hilbert space are either equivalent or mutually singular. We will briefly state the whole chain of reasoning, and then provide the necessary details.

We will show that our measure at all times is Gaussian. By the Feldman-Hájek theorem, two Gaussian measures $\mathcal{N}(\mathbf{m}_1, \mathbf{C}_1)$ and $\mathcal{N}(\mathbf{m}_2, \mathbf{C}_2)$ on a Hilbert space are either equivalent or mutually singular. A necessary condition for equivalence is that the Cameron-Martin spaces, as given by $\mathbf{C}^{1/2}$, of the two measures are equal as sets Da Prato & Zabczyk (1992)[Theorem 2.23]. Thus, if the ranges of the covariance operators $\mathbf{C}_1^{1/2}$ and $\mathbf{C}_2^{1/2}$ are not equal, then the measures are mutually singular and their TV distance is 1.

**Calculating the Covariance Operators** Starting with a measure $\mu_0$ of the initial distribution, the heat equation induces a semigroup $S(t) = \exp(t\mathbf{\Lambda})$, which maps the initial condition $u_0$ to the solution at time $t$, i.e. $u(\cdot, t) = S(t)u_0$. Therefore, the measure $\mu_t$ of $u(\cdot, t)$ is induced by the pushforward measure $\mu_0$ under $S(t)$, i.e. $\mu_t = S(t)_\# \mu_0$. As $S(t)$ is linear, $\mu_t$ is also a Gaussian measure with mean $\mathbf{0}$ and covariance operator

$$\mathbf{C}_t = S(t)(\mathbf{I} - \mathbf{\Lambda})^{-1}S(t)^* = \exp(t\mathbf{\Lambda})(\mathbf{I} - \mathbf{\Lambda})^{-1}\exp(t\mathbf{\Lambda}),$$

where $S(t)^*$ denotes the adjoint of $S(t)$ (Hairer (2023) Chap. 4.3). As we have seen above, the eigenvalues of $(\mathbf{I} - \mathbf{\Lambda})^{-1}$ are given by $\mu_n = 1/(1 + (n\pi)^2)$, with corresponding eigenfunctions

$e_n(x)$. Further, the eigenfunctions of $\boldsymbol{\Lambda}$ are also given by $e_n(x)$, with corresponding eigenvalues $\lambda_n = -(n\pi)^2$. Lastly, by functional calculus, the eigenfunctions of $\exp(t\boldsymbol{\Lambda})$ are also given by $e_n(x)$ with corresponding eigenvalues $\nu_n = \exp(-t(n\pi)^2)$. With this, we can compute

$$\boldsymbol{C}_t \boldsymbol{e}_n = \exp(t\boldsymbol{\Lambda})(\boldsymbol{I} - \boldsymbol{\Lambda})^{-1}\exp(t\boldsymbol{\Lambda})\boldsymbol{e}_n$$

$$= (\exp(-t(n\pi)^2))\left(\frac{1}{1+(n\pi)^2}\right)(\exp(-t(n\pi)^2))\boldsymbol{e}_n = \frac{\exp(-2t(n\pi)^2)}{1+(n\pi)^2}\boldsymbol{e}_n.$$

Thus, the eigenvalues of $\boldsymbol{C}_t$ are given by $\lambda_n(t) = \nu_n^2 \mu_n = \exp(-2t(n\pi)^2)/(1+(n\pi)^2)$, with corresponding eigenfunctions $e_n(x)$.

**Calculating the Cameron-Martin Spaces** The functions $e_n(x)$ form an orthonormal basis of $L^2(\Omega)$, so we can express every element $f \in L^2(\Omega)$ as

$$f = \sum_{n=1}^{\infty} c_n e_n, \quad \sum_{n=1}^{\infty} c_n^2 < \infty.$$

The Cameron-Martin space $H_t$ of $\mathcal{P}_t$ is given by the range of $\boldsymbol{C}_t^{1/2}$, which is given by

$$\mathrm{Ran}(\boldsymbol{C}_t^{1/2}) = \left\{\boldsymbol{C}_t^{1/2}f \mid f \in L^2(\Omega)\right\} = \left\{\sum_{n=1}^{\infty}\sqrt{\lambda_n(t)}c_n e_n \mid f \in L^2(\Omega)\right\}.$$

Therefore, $g \in H_t$ if and only if $g$ can be expressed as

$$g = \sum_{n=1}^{\infty} d_n e_n, \quad \sum_{n=1}^{\infty}\frac{d_n^2}{\lambda_n(t)} < \infty.$$

Inserting the expression for $\lambda_n(t)$, we see that $g \in H_t$ if and only if

$$\sum_{n=1}^{\infty} d_n^2 \frac{1+(n\pi)^2}{\exp(-2t(n\pi)^2)} < \infty.$$

**Showing That the Cameron-Martin Spaces Are Not Equal** Now it is easy to see that for any $t \geq 0, \delta > 0$, the Cameron-Martin spaces $H_t$ and $H_{t+\delta}$ are not equal. For example, the function

$$h(x) = \sum_{n=1}^{\infty}\exp(-(t+\delta)(n\pi)^2)e_n(x)$$

is an element of $H_t$, as

$$\sum_{n=1}^{\infty}(\exp(-(t+\delta)(n\pi)^2))^2\frac{1+(n\pi)^2}{\exp(-2t(n\pi)^2)} = \sum_{n=1}^{\infty}(1+(n\pi)^2)\exp(-2\delta(n\pi)^2) < \infty,$$

but it is not an element of $H_{t+\delta}$, as

$$\sum_{n=1}^{\infty}(\exp(-(t+\delta)(n\pi)^2))^2\frac{1+(n\pi)^2}{\exp(-2(t+\delta)(n\pi)^2)} = \sum_{n=1}^{\infty}(1+(n\pi)^2) = \infty.$$

Therefore, by the Feldman-Hájek theorem, the measures $\mathcal{P}_t$ and $\mathcal{P}_{t+\delta}$ are mutually singular, and their TV distance is 1. $\qquad\square$

A.3 ILLUSTRATION OF THE METHOD OF LINES

Consider the one-dimensional heat equation on the domain $\Omega = (0,1)$ with Dirichlet boundary conditions

$$\begin{aligned}\frac{\partial u}{\partial t}(x,t) &= \frac{\partial^2 u}{\partial x^2}(x,t), \quad x \in (0,1), \, t \geq 0,\\ u(0,t) = u(1,t) &= 0, \quad t \geq 0,\\ u(x,0) &= u_0(x), \quad x \in (0,1),\end{aligned}$$

where $u : \bar{\Omega} \times [0,\infty) \to \mathbb{R}$ denotes the temperature field. We will numerically solve this PDE using the *method of lines*.

**Method of Lines** We discretize the spatial domain with a uniform grid $\mathcal{M} = \{x_1, \ldots, x_n\} \subset \Omega$ with $x_i = \frac{i}{n+1}, i \in \{1, \ldots, n\}$, while leaving the time domain continuous. We can approximate the second derivative in space with the finite difference scheme

$$\frac{\partial^2 u}{\partial x^2}(x_i, t) \approx \frac{u(x_{i+1}, t) - 2u(x_i, t) + u(x_{i-1}, t)}{(\Delta x)^2}, \quad \Delta x = \frac{1}{n+1}.$$

This leads to the system of ODEs.

$$\frac{d\tilde{\boldsymbol{u}}(t)}{dt} = \boldsymbol{A}\tilde{\boldsymbol{u}}(t),$$

where $\boldsymbol{A} \in \mathbb{R}^{n \times n}$ is the matrix

$$\boldsymbol{A} := \frac{1}{(n+1)^2} \begin{pmatrix} -2 & 1 & 0 & \cdots & 0 \\ 1 & -2 & 1 & \cdots & 0 \\ 0 & 1 & -2 & \cdots & 0 \\ \vdots & \vdots & \vdots & \ddots & \vdots \\ 0 & 0 & 0 & \cdots & -2 \end{pmatrix},$$

and $\tilde{\boldsymbol{u}}(t) \in \mathbb{R}^n$ is the discretization of $u(\cdot, t)$ on the grid $\mathcal{M}$. Consequently, the solution to this system of ODEs can be expressed in terms of the matrix exponential $\tilde{\boldsymbol{u}}(t) = \exp(t\boldsymbol{A})\,\tilde{\boldsymbol{u}}(0)$.

## A.4 TV Distance Bound

**Theorem A.2.** *Let $\mathcal{P}_t, \mathcal{P}_{t+\delta}$ be the laws of $u_t, u_{t+\delta}$. Under the assumptions and with the notation from theorem 4.2,*

$$d_{\mathrm{TV}}(\mathcal{P}_t, \mathcal{P}_{t+\delta}) = \tfrac{1}{2} \int_{\mathbb{R}^n} |p_t(\boldsymbol{x}) - p_{t+\delta}(\boldsymbol{x})| \, d\boldsymbol{x}, \tag{2}$$

*where $p_t, p_{t+\delta}$ are the densities of $\mathcal{P}_t, \mathcal{P}_{t+\delta}$, and*

$$d_{\mathrm{TV}}(\mathcal{P}_t, \mathcal{P}_{t+\delta}) \le \sqrt{\frac{1}{4}\Big[tr(\boldsymbol{\Sigma}_{t+\delta}^{-1}\boldsymbol{\Sigma}_t) - n + (\Delta\mu)^T\boldsymbol{\Sigma}_{t+\delta}^{-1}\Delta\mu + \log\frac{\det(\boldsymbol{\Sigma}_{t+\delta})}{\det(\boldsymbol{\Sigma}_t)}\Big]}, \tag{3}$$

*where*

$$\Delta\mu = (\boldsymbol{\mu}_{t+\delta} - \boldsymbol{\mu}_t) \quad \boldsymbol{\mu}_t = \exp(t\boldsymbol{A})\boldsymbol{\mu}_0 + \int_0^t \exp((t-s)\boldsymbol{A})\boldsymbol{r}(s)\mathrm{d}s, \quad \boldsymbol{\Sigma}_t = \exp(t\boldsymbol{A})\boldsymbol{\Sigma}_0\exp(t\boldsymbol{A}^T).$$

*Proof.* We know that $\mathcal{P}_t, \mathcal{P}_{t+\delta}$ admit densities from theorem 4.2. 2 follows by the definition of the TV distance between two distributions with densities $p_t, p_{t+\delta}$. For 3, we use Pinsker's inequality, which yields an upper bound on the TV distance by the Kullback–Leibler divergence

$$d_{\mathrm{TV}}(\mathcal{N}(\boldsymbol{\mu}_1, \boldsymbol{\Sigma}_1), \mathcal{N}(\boldsymbol{\mu}_2, \boldsymbol{\Sigma}_2)) \le \sqrt{\frac{1}{2}D_{\mathrm{KL}}(\mathcal{N}(\boldsymbol{\mu}_1, \boldsymbol{\Sigma}_1) \,\|\, \mathcal{N}(\boldsymbol{\mu}_2, \boldsymbol{\Sigma}_2))}.$$

The Kullback–Leibler divergence of two Gaussians above is well known, and given by

$$\frac{1}{2}\left[tr(\boldsymbol{\Sigma}_2^{-1}\boldsymbol{\Sigma}_1) - n + (\boldsymbol{\mu}_2 - \boldsymbol{\mu}_1)^T\boldsymbol{\Sigma}_2^{-1}(\boldsymbol{\mu}_2 - \boldsymbol{\mu}_1) + \log\frac{\det(\boldsymbol{\Sigma}_2)}{\det(\boldsymbol{\Sigma}_1)}\right].$$

Thus the claim follows by plugging in the calculated means and covariances. $\square$

## A.5 Data Generation

We generated multiple synthetic datasets from a general second order formulation for periodic PDEs, each discretized with finite-difference schemes. Initial conditions were sampled from Gaussian processes with covariance $(-\partial_x^2 + 25I)^{-2}$ to provide smooth but nontrivial trajectories. Each dataset

is stored as a compressed `.npz` file containing $(n_{\text{samples}}, n_t + 1, n_x)$ trajectories together with grid and metadata. The PDE looks as follows

$$u_t(x,t) + au(x,t)_{xx} + bu(x,t)_x + cu(x,t) = 0, \qquad x \in (0,1), \ t \geq 0,$$
$$u(0,t) = u(1,t) = 0, \qquad t \geq 0,$$
$$u(x,0) = u_0(x), \quad x \in (0,1).$$

where $a, b, c \in \mathbb{R}$ are tuneable parameters. We sampled data from all combinations of $a \in \{-0.005, -0.0075, -0.01\}$, $b \in \{-0.5, 0, 0.5\}$, and $c \in \{-0.5, 0, 0.5\}$. Note that we focused on negative values for $a$, as positive values tend to smoothen the trajectory over time which leads to trivial coverage even for naïve CP. We sampled 21 time steps (incl. initial condition) between 0 and 1.

### A.6 REAL-WORLD EXAMPLE

The dataset of Wei et al. (2023) contains 19 samples of 2D surfaces measured over 1810 time steps. We allocate 8 samples for training a Fourier Neural Operator Li et al. (2021), 1 for validation, 7 for calibration, and 3 for testing. Calibration begins at the end of the heating process, and testing proceeds over the subsequent time steps. We crop the borders of the measurement data, since they correspond to non-heated background regions (see figure 4). We use outlier clipping and a Gaussian filter to get rid of the measurement noise, and normalize the data with values of the first time step. Because the dataset is small, we divide each surface into square patches of size $5 \times 5$ and treat them as identically distributed samples (to mitigate dependency between the patches, we leave an empty space of size 5 to all neighboring patches). While this induces some dependence between patches, and the variations in the metal plates (e.g., differently drilled backside holes) introduce slight distribution shifts, both effects are minor for a real-world measurement dataset and have negligible impact on CP performance. As the system cools, both the data and prediction residuals smooth out over time, causing many CP methods to overcover rather than undercover. We show the results in figure 5. Weighted CP, which only detects that the distribution has changed, still produces infinite intervals. Despite this limitation and the relative advantage of standard CP in this specific setting, our experiment shows that the weighted CP approach remains practically applicable.

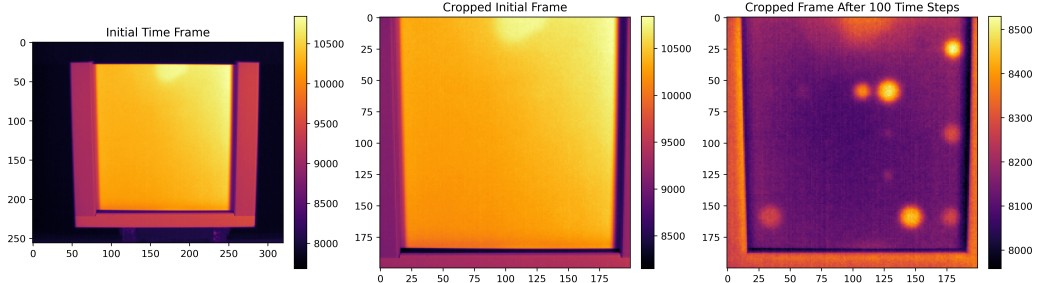

Figure 4: Exemplary frame from the real-world heat pulsed-thermography dataset. The left image shows a raw data frame from the beginning of the cooling period. The middle image shows how we cropped the frames to remove the background. The right image shows how the data looks after 100 time steps.

### A.7 ADDITIONAL RESULTS ON SYNTHETIC DATA

We show results for remaining $b$-values in figures 6 and 7. For $b = -0.5$, we report the remaining numbers in tables 2 and 3. In Figure 8, we see an example of CP bands where naïve CP and LSCI undercover and WCP remains full coverage.

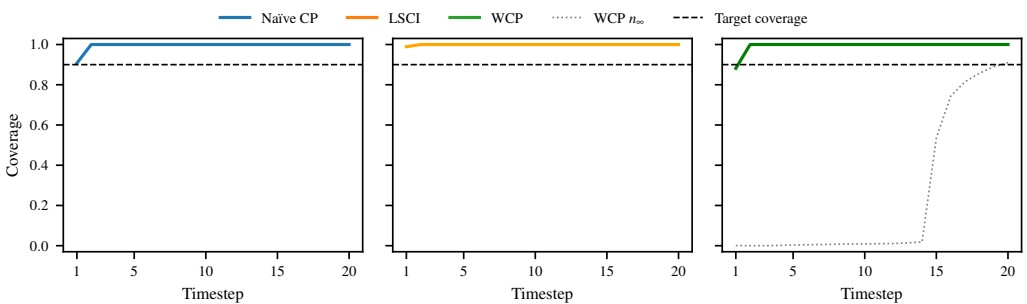

Figure 5: Coverages of the CP methods on the heat pulsed-thermography dataset. As the data smoothens out over time, coverages become trivial at some point.

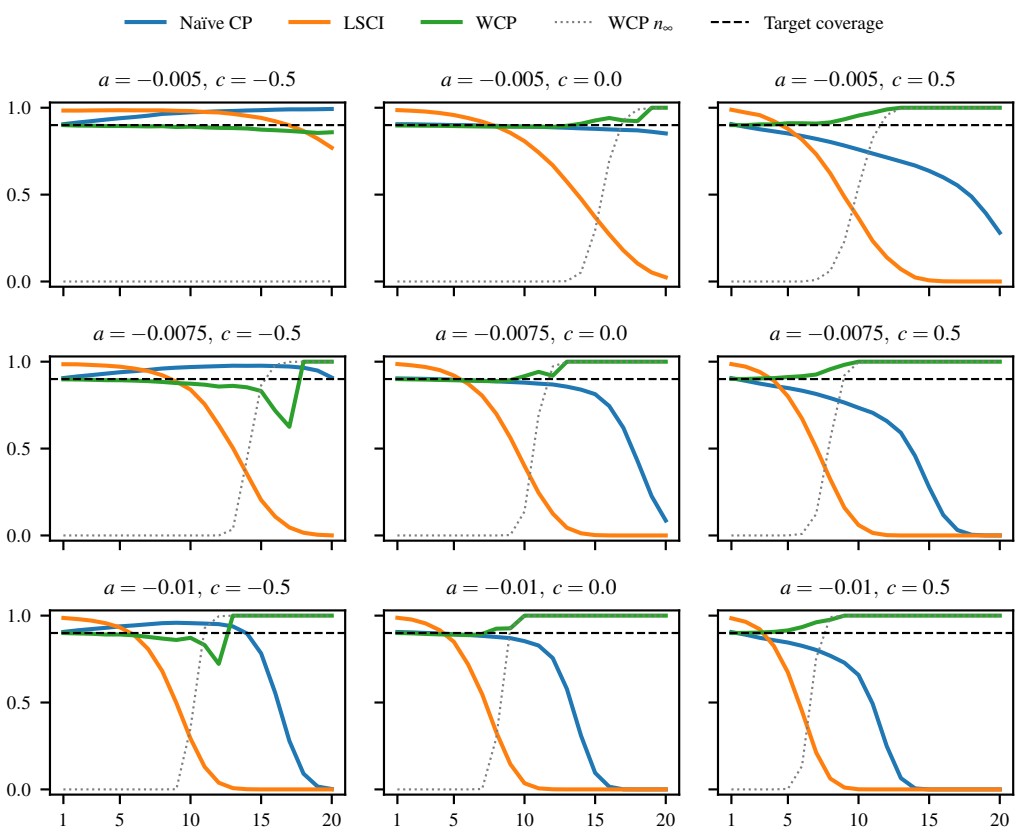

Figure 6: Mean coverages for varying $a, c$ and fixed $b = 0$ across increasing prediction horizon. We omit coverages when infinite conformal bands were reported (coverage of 1 would hold trivially). The 90% coverage target and the fraction of samples with infinite bandwidth $n_\infty$ are marked as black dashed line and gray dotted line, respectively.

## A.8    OTHER INITIAL DISTRIBUTIONS

To show that our method also works when sampling data from other distributions of the location-scale family, we include coverage statistics when using the Laplace and the logistic distribution for the initial values (only for $b = -0.5$, to keep this outlook concise). The figure for Laplace can be found in 9 and for logistic in 10.

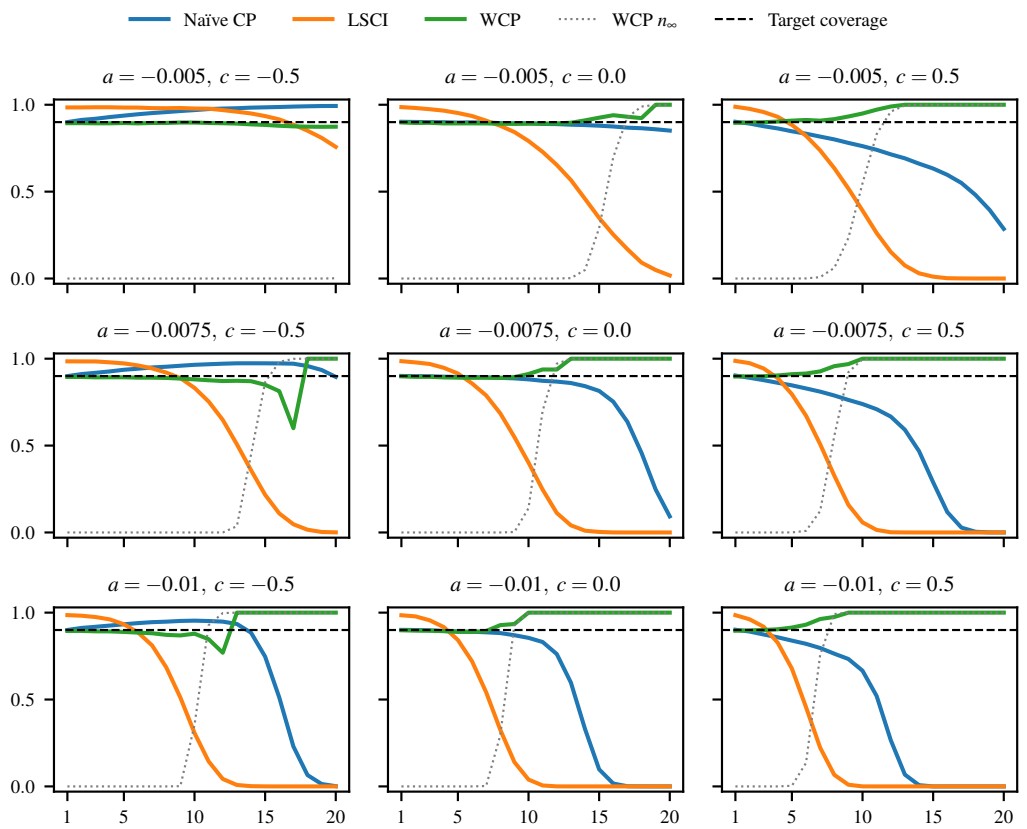

Figure 7: Mean coverages for varying $a, c$ and fixed $b = 0.5$ across increasing prediction horizon. We omit coverages when infinite conformal bands were reported (coverage of 1 would hold trivially). The 90% coverage target and the fraction of samples with infinite bandwidth $n_\infty$ are marked as black dashed line and gray dotted line, respectively.

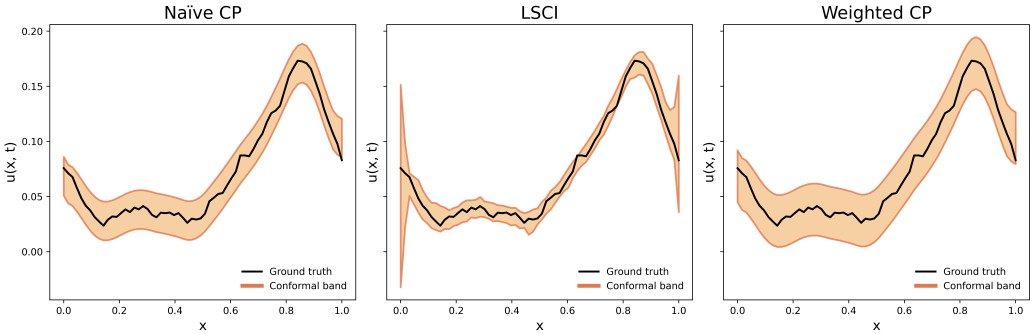

Figure 8: Exemplary coverage plots for the three methods naïve CP (left), LSCI (middle), and WCP (right) for a sample at time step 7 with parameters $a = -0.005, b = 0.5, c = 0.5$. In this case, only WCP fully covers the the trajectory.

## A.9   USE OF LARGE LANGUAGE MODELS

Large language models (LLMs) helped in the creation and execution of this project. They assisted with improving the clarity and readability of the manuscript, suggesting alternative phrasings, providing feedback on mathematical arguments, and offering ideas during the research and coding process. All research contributions, results, and final formulations were verified manually.

Table 2: Mean coverages and bandwidths over 5000 sampled initial conditions for varying $a$ and fixed $b = -0.5$, $c = 0$. For WCP, we also report the fraction of samples where infinite bands were reported ($n_\infty$) to maintain coverage guarantees. The gray font is chosen for better readability.

| | | | | Timestep | | | |
|---|---|---|---|---|---|---|---|
| | | | 1 | 5 | 10 | 15 | 20 |
| $a = -0.005$ | Naïve CP | Coverage | 0.9 | 0.9 | 0.9 | 0.88 | 0.85 |
| | | Bandwidth | 0.03 | 0.03 | 0.03 | 0.03 | 0.03 |
| | LSCI | Coverage | 0.99 | 0.96 | 0.82 | 0.39 | 0.03 |
| | | Bandwidth | 0.02 | 0.02 | 0.02 | 0.02 | 0.02 |
| | WCP (Ours) | Coverage | 0.9 | 0.9 | 0.9 | 0.93 | 1.0 |
| | | Bandwidth | 0.03 | 0.03 | 0.03 | 0.04 | 0.04 |
| | | $n_\infty$ | 0.0% | 0.0% | 0.0% | 29.4% | 100.0% |
| $a = -0.0075$ | Naïve CP | Coverage | 0.9 | 0.9 | 0.88 | 0.8 | 0.08 |
| | | Bandwidth | 0.03 | 0.03 | 0.03 | 0.03 | 0.03 |
| | LSCI | Coverage | 0.99 | 0.92 | 0.43 | 0.0 | 0.0 |
| | | Bandwidth | 0.02 | 0.02 | 0.02 | 0.02 | 0.02 |
| | WCP (Ours) | Coverage | 0.9 | 0.9 | 0.92 | 1.0 | 1.0 |
| | | Bandwidth | 0.03 | 0.03 | 0.04 | $\infty$ | $\infty$ |
| | | $n_\infty$ | 0.0% | 0.0% | 14.0% | 100% | 100% |
| $a = -0.01$ | Naïve CP | Coverage | 0.91 | 0.9 | 0.86 | 0.09 | 0.0 |
| | | Bandwidth | 0.03 | 0.03 | 0.03 | 0.03 | 0.03 |
| | LSCI | Coverage | 0.99 | 0.86 | 0.04 | 0.0 | 0.0 |
| | | Bandwidth | 0.02 | 0.02 | 0.02 | 0.02 | 0.02 |
| | WCP (Ours) | Coverage | 0.9 | 0.9 | 1.0 | 1.0 | 1.0 |
| | | Bandwidth | 0.03 | 0.03 | 0.07 | $\infty$ | $\infty$ |
| | | $n_\infty$ | 0.0% | 0.0% | 99.9% | 100% | 100% |

Table 3: Mean coverages and bandwidths over 5000 sampled initial conditions for varying $a$ and fixed $b = -0.5$, $c = 0.5$. For WCP, we also report the fraction of samples where infinite bands were reported ($n_\infty$) to maintain coverage guarantees. The gray font is chosen for better readability.

| | | | Timestep | | | | |
|---|---|---|---|---|---|---|---|
| | | | 1 | 5 | 10 | 15 | 20 |
| $a = -0.005$ | Naïve CP | Coverage | 0.9 | 0.85 | 0.76 | 0.63 | 0.27 |
| | | Bandwidth | 0.03 | 0.03 | 0.03 | 0.03 | 0.03 |
| | LSCI | Coverage | 0.99 | 0.89 | 0.4 | 0.01 | 0.0 |
| | | Bandwidth | 0.02 | 0.02 | 0.02 | 0.02 | 0.02 |
| | WCP (Ours) | Coverage | 0.9 | 0.91 | 0.95 | 1.0 | 1.0 |
| | | Bandwidth | 0.03 | 0.04 | 0.05 | 0.07 | $\infty$ |
| | | $n_\infty$ | 0.0% | 0.0% | 55.1% | 100.0% | 100% |
| $a = -0.0075$ | Naïve CP | Coverage | 0.91 | 0.85 | 0.74 | 0.26 | 0.0 |
| | | Bandwidth | 0.03 | 0.03 | 0.03 | 0.03 | 0.03 |
| | LSCI | Coverage | 0.99 | 0.82 | 0.07 | 0.0 | 0.0 |
| | | Bandwidth | 0.02 | 0.02 | 0.02 | 0.02 | 0.02 |
| | WCP (Ours) | Coverage | 0.9 | 0.91 | 1.0 | 1.0 | 1.0 |
| | | Bandwidth | 0.03 | 0.04 | 0.06 | $\infty$ | $\infty$ |
| | | $n_\infty$ | 0.0% | 0.0% | 99.7% | 100% | 100% |
| $a = -0.01$ | Naïve CP | Coverage | 0.9 | 0.85 | 0.65 | 0.0 | 0.0 |
| | | Bandwidth | 0.03 | 0.03 | 0.03 | 0.03 | 0.03 |
| | LSCI | Coverage | 0.99 | 0.71 | 0.0 | 0.0 | 0.0 |
| | | Bandwidth | 0.02 | 0.02 | 0.02 | 0.02 | 0.02 |
| | WCP (Ours) | Coverage | 0.9 | 0.91 | 1.0 | 1.0 | 1.0 |
| | | Bandwidth | 0.03 | 0.04 | $\infty$ | $\infty$ | $\infty$ |
| | | $n_\infty$ | 0.0% | 0.3% | 100% | 100% | 100% |

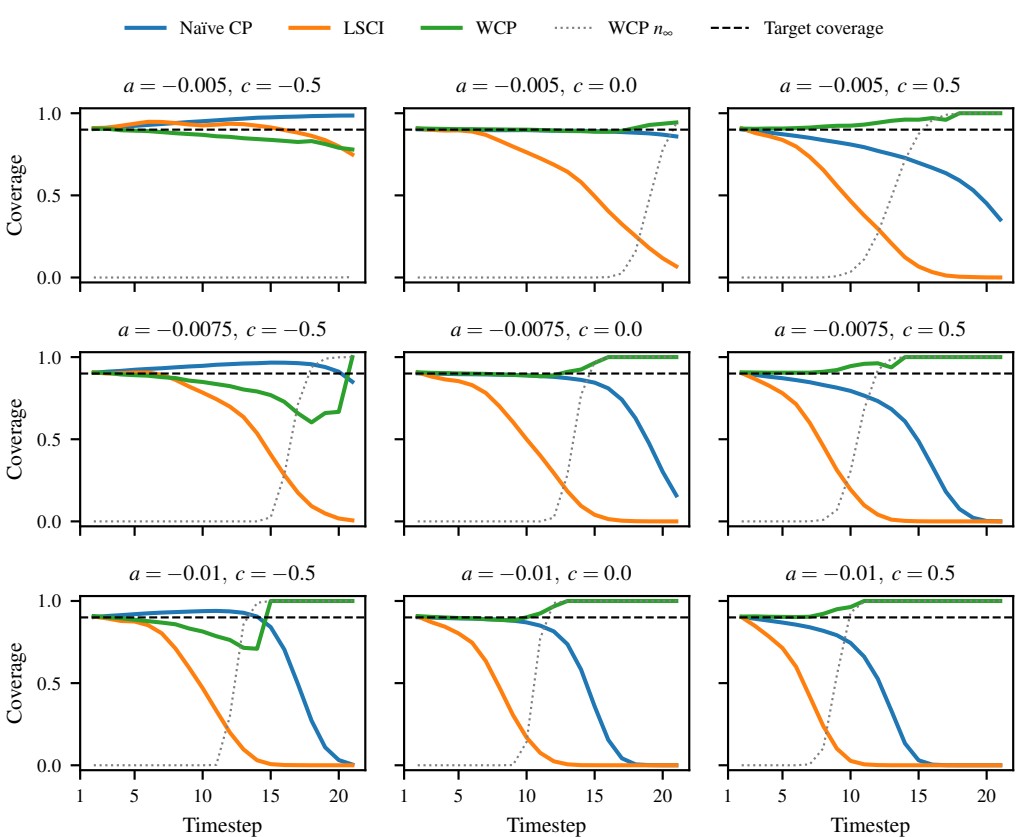

Figure 9: Mean coverages when sampling from a Laplacian initial distribution for varying $a, c$ and fixed $b = -0.5$ across increasing prediction horizon. We omit coverages when infinite conformal bands were reported (coverage of 1 would hold trivially). The 90% coverage target and the fraction of samples with infinite bandwidth $n_\infty$ are marked as black dashed line and gray dotted line, respectively.

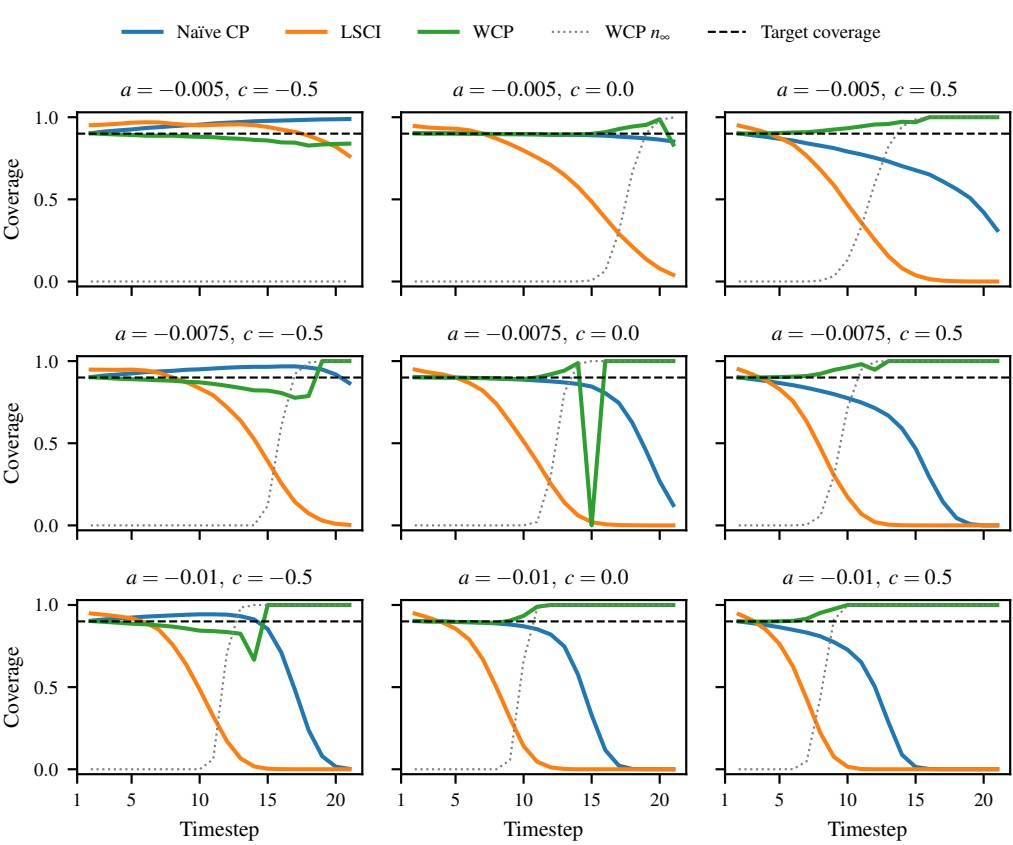

Figure 10: Mean coverages when sampling from a logistic initial distribution for varying $a, c$ and fixed $b = -0.5$ across increasing prediction horizon. We omit coverages when infinite conformal bands were reported (coverage of 1 would hold trivially). The 90% coverage target and the fraction of samples with infinite bandwidth $n_\infty$ are marked as black dashed line and gray dotted line, respectively.

