# OpenReview forum: "Weighted Conformal Prediction for Time-Dependent PDEs"
_ICLR.cc/2026/Conference — Submitted to ICLR 2026_

### Official Review · Reviewer_iHz7 · 2025-10-20

**Soundness:** 3
**Presentation:** 3
**Contribution:** 1
**Rating:** 2
**Confidence:** 4

**Summary:**

Spatiotemporal PDEs are being increasingly solved with neural operators. Uncertainty quantification on such predictions, therefore, is not immediately possible as a result of distribution shifts that occur as temporal windows slide. In this vein, previous works have studied conformal prediction under covariate shift. This paper then studied the application of such ideas to this setting, first in the full functional form and then in the discretized setting.

**Strengths:**

The paper presents an interesting problem of study: the study of spatiotemporal PDE coverage seems like a worthwhile direction. The exposition of the paper is also very clear, making the paper easy to follow and engage with. The paper also nicely discusses both the functional and the discretized forms of the problem statement.

**Weaknesses:**

In the first part of the paper, the authors establish that the TV distance between consecutive steps is maximal in the functional space. I believe the implication of this statement is supposed to be that, therefore, we cannot consider the functional space coverage using ideas from covariate shift. However, the approach of Barbet et. al only establishes that $\mathcal{P}(Y\in\mathcal{C}(X))\ge 1-\alpha-\sum_i w_i d_{TV}(z, z^i)$, which is a lower bound. How can we say anything more strongly than just making claims of this lower bound here?

Moreover, this then motivates studying this problem in a finite-dimensional discretized form. This, however, then just becomes a standard, finite-dimensional time series problem, so I do not see how this becomes any different from a typical finite-dimensional conformal time series problem. Also, Theorem 4.2 appears to be a standard proof from SDEs, so I do not see why it and its proof are provided in the main body of the paper.

Finally, the experimental results appear to be not very compelling: many times, WCP appears to produce extremely conservative prediction regions, having a coverage of 1.0 or close to it at desired coverage levels well below this. The other methods, in contrast, appropriately hit 0.90 initially.

**Questions:**

1. How does this particular setting differ from a general time series conformal prediction problem after the PDE has been discretized? Are there particular structures that you can uniquely leverage to develop a method in this setting that cannot be more generally applied? This reduced scope would likely yield more useful methods.

2. Why are the coverages so conservative in this setting? The empirical results seem to indicate over-coverage even before the time steps progress very far.

3. Even though the TV in the functional case is 1.0, does that necessarily mean there is no other form of covariate-shift type adjustments that could fix the miscoverage?

---

> ### Author Response · Authors · 2025-11-21
>
> Thank you very much for review! You highlighted some very interesting questions that we made sure to address in our updated version.
>
> We first want to mention that we have significantly reworked our experiment section, which previously was driven by poor design choices and a bug in the code. Instead of hand-picked specific PDEs, we switched to general second-order PDEs with tunable parameters, which cover the most used PDEs in specific parameter combinations.
> We also added a 2D real-world example, showing that our method can be readily applied in practice.
>
> Regarding your questions:
>
> 1. You are right to observe that this is a time series setting. However, time series problems remain largely unsolved for conformal prediction. Most existing methods provide only asymptotic coverage guarantees. This means they may undercover at one time step and overcover at another, typically showing that these miscoverages vanish only over an infinite time horizon.
>    You are also correct that we have identified additional structure that we can exploit. Our contribution is precisely the identification of linear PDEs as a class where such structure exists, together with a detailed analysis of when and how it can be used. Concretely, we show that in a discretized setting the initial distributions evolve in a predictable manner, which allows us to apply the Weighted Conformal Prediction method of Barber et al. We also demonstrate that this is not possible when considering the functional formulation of the problem.
>    We are therefore the first to achieve per-times step coverage guarantees for PDEs. While we can only proof this for linear PDEs, we believe that our work is an important starting point and relevant contribution to the community.
> 2. The method overcovers by so much because of a poor choice in the evaluation design. The LSCI paper uses the convention that a single sample is considered covered when at least 99% of its points are covered, and we kept this for a fair comparison. However, we also find this convention confusing and unnecessary, and it was the reason our method initially overcovered. In the revised version, we removed this distinction and now consider a sample covered only if all of its points are covered. We increased the sampling rate in LSCI to compensate for this change.
> 3. This is a very interesting question, and the answer has several facets.
>
> 	1. Yes, this is only a lower bound, and there may be datasets that achieve good empirical coverage. The problem is that we lose theoretical guarantees, so we cannot know in advance whether a dataset will be covered. This defeats the purpose of conformal prediction.
> 	2. It is indeed a direct consequence of the singularity between the calibration and test distributions that regular CP, as well as many downstream extensions such as weighted CP or adaptive CP, are not applicable. Conformal prediction fundamentally relies on the calibration and test distributions being the same (a consequence of exchangeability). Some methods allow a controlled difference between them, with the final fallback being a total variation bound, but none of these apply when calibration and test distributions are mutually singular.
> 	3. The main point we wanted to convey is slightly different. Several of the papers we cited use the heuristic that with a small enough time step the distribution changes only marginally, so standard conformal prediction remains valid. We wanted to emphasize that this heuristic is fundamentally flawed.
> 	4. Finally, as you state, there may exist methods that achieve guarantees even in this setting, but they would need to exploit strong structural biases of the underlying PDEs. So far, we are not aware of any such method.
> 	Thank you again for raising this question. We have added further clarification in the text text after the theorem to highlight these consequences.
>
> We acknowledge that, especially given the limited experiment section in the initial manuscript, the paper lacked practical value. We hope that the revised experiment section, together with our discussion of the current state of methods that offer conformal guarantees, addresses this concern and convinces you to recommend the paper for publication.

---

> ### Comment · Reviewer_iHz7 · 2025-11-27
>
> I thank the authors for providing this rebuttal. Regarding the points:
>
> 1. Thank you for pointing this out: I now see that any SPDE evolution for which there is a density form can be used in this WCP formulation (since you need to have the densities to appropriately do the reweighting). I believe this idea can be extended beyond the linear class, but this contribution is now clear to me.
>
> 2. Practically, it still seems that this approach fails to be very practically useful, as after some point, the conformal region covers nearly the full space. The alternatives become similarly vacuous (in the opposite direction) by achieving 0% coverage; however, it is not clear if one is really preferable to the other for practical purposes. The only case where this is not true is for alpha = 0.005, but in most practical settings, we are often interested in much large values of alpha, like 0.01 or even 0.05, where it seems this method greatly overcovers the truth.
>
> 3. I see -- the point that this method seeks to extend beyond the assumption of "small enough time step the distribution changes only marginally, so standard conformal prediction remains valid" should be emphasized more clearly (motivating the correction leveraged in part 1).
>
> My overall read on the paper is more positive than my original reading (and have raised my score accordingly); however, I do still think there is more work necessary for publication. The work does identify a class of problems (in particular PDEs) of time series forecasting where distribution shifts can actually be anticipated by virtue of the analytic form of the density evolution being known (up to normalization), unlike a generic time series setting. However, a couple questions still exist:
>
> 1. It is still not practically clear when leveraging this structure is actually significant: the experiments are still not especially convincing of this method improving much over standard CP. Ideally, one will find PDE settings where the standard CP coverage becomes vacuous (either over covering or completely undercovering), where WCP in contrast achieves genuinely useful guarantees. It seems almost certainly the case that such settings exist (even in the restricted domain of linear PDEs).
>
> 2. Given that we assume the form of the density is known, why are we doing this calibration with CP at all? The whole point of CP is to achieve coverage *without* distributional assumptions. Standard covariate shift usually does *not* assume the density is known and instead estimates this likelihood ratio by training a separate classification model.

---

> > ### Author Response · Authors · 2025-11-28
> >
> > Thank you very much for the time and effort to go over the submission again and consider our answers. We appreciate that you decided to raise the score.
> >
> > We haven’t considered extending our work to SPDEs, but this sounds very interesting and we will definitely consider this in future work.
> >
> > We want to use the opportunity to address your last questions (also for other readers of this discussion):
> >
> > 1. **Why would vacuous overcoverage be preferable to undercoverage?**
> > We would strongly argue that a machine that confidently provides the wrong answer is worse than a machine that refrains from answering (or admits lack of confidence through vacuous bounds), because in the latter case human intervention could still prevent failure.
> > 2. **Why are we doing CP if we already know the density?**
> > We only know the density at time step (t−1), but have no information about the density at time step (t). We need a model to predict values for time step (t), and we want to have an accurate error estimation for the model’s prediction. This is why we need this variant of CP.
> >
> > We further want to highlight that the reviewer is correct that standard conformal prediction is distribution-free. However, standard CP is also limited (e.g., not applicable to time-series), which is why there are variants of CP that assume additional information from the data.
> >
> > Thank you again for your high-quality review and the constructive discussion!

---

### Official Review · Reviewer_e72z · 2025-10-29

**Soundness:** 3
**Presentation:** 2
**Contribution:** 1
**Rating:** 2
**Confidence:** 3

**Summary:**

This paper develops weighted conformal prediction (WCP)  for applying conformal prediction to time-dependent PDE surrogate models.  The authors prove that in function spaces with time-dependency, distributions at different times can be mutually singular (TV distance = 1), breaking the exchangeability assumption and hence CP coverage guarantees. As a solution, they propose reweighing the nonconformity scores with density ratios to enable exact coverage guarantees. Experiments on three PDEs (univariate, linear, with Gaussian initial conditions) demonstrate that WCP maintains coverage where two baselines, naive CP and LSCI, fail.

**Strengths:**

- The problem is significant and well motivated. The authors make a good point in laying out the challenge of covariance shifts in time-dependent PDEs that are common in physics and engineering. Theorem 4.1 was also a clear illustration of this challenge and how existing methods can fail.

- The paper was over-all well organized and easy to follow.

**Weaknesses:**

- No real algorithmic or theoretical contribution. Theorem 4.1 proves that for the heat equation with Gaussian initial conditions, the TV distance between solution distributions at any two distinct time points is maximal (equals 1). Theorem 4.2 shows that for discretized linear PDEs with Gaussian initial conditions, the solution distribution at any time t remains Gaussian, with explicitly computable mean and covariance. These two results are not particular novel for PDEs. The authors' method, then, directly applies Barber et al. 2023's weighted conformal prediction algorithm using the likelihoods. The coverage guarantees are from Barber 2023. I fail to see how this algorithm is a novel contribution.

- The scope is limited. The algorithm only applies to _linear_ PDEs with _Gaussian_ (and location-scale) initial conditions, but most physical systems of interest of UQ have nonlinear dynamics (Navier-Stokes, reaction-diffusion with nonlinear terms, etc.). The proposed algorithm require explicit probabilities which makes it not extendable to the more general case.

- Significant weakness and ambiguity in the experiment section. This includes:
    - What is the target coverage? Why is the target in Fig 3 set at 90% but the target for WCP said to be 99%? Why does WCP's line disappear half way through the horizon? What is $n_\infty$ in Table 1 and how should we interpret it?
    - Limited experiment setups: only linear, univariate, synthetic experiments. This is a result of the nature and assumption made by the method, but with this set of experiments it's not very convincing that this algorithm is useful in practice.
    - Lack of comparison to baselines, and discussion of literature. From the PDE literature, there are works that provide probabilistic forecasts (for example [1,2], and the baselines in LSCI). From the CP literature, there are a swath of works that handles time series distribution shift that does not require explicit covariates or local exchangeability [3-6]. They should be compared to show what are the advantages of your algorithm, or at least discussed w.r.t. why they are not applicable to your setup.
    - It is also questionable that WCP over-covers by so much, often achieving 1.0 coverage with infinite bands. While I respect the authors' honest reporting, this is a significant limitation on the usability of the UQ bands. UQ's goal is calibration and the tendency to over-cover is not desirable. Can you show results of both 90\% and 99\% target coverage?
    - Some qualitative plots of UQ bands (similar to figure 1) will also help the presentation of results.

[1] Yang, Liu, Xuhui Meng, and George Em Karniadakis. "B-PINNs: Bayesian physics-informed neural networks for forward and inverse PDE problems with noisy data." Journal of Computational Physics 425 (2021): 109913.

[2] Bülte, Christopher, Philipp Scholl, and Gitta Kutyniok. "Probabilistic neural operators for functional uncertainty quantification." arXiv preprint arXiv:2502.12902 (2025).

[3] Gibbs, Isaac, and Emmanuel Candes. "Adaptive conformal inference under distribution shift." Advances in Neural Information Processing Systems 34 (2021): 1660-1672

[4] Angelopoulos, Anastasios, Emmanuel Candes, and Ryan J. Tibshirani. "Conformal pid control for time series prediction." Advances in neural information processing systems 36 (2023): 23047-23074.

[5] Xu, Chen, and Yao Xie. "Sequential predictive conformal inference for time series." International Conference on Machine Learning. PMLR, 2023.

[6] Auer, Andreas, et al. "Conformal prediction for time series with modern hopfield networks." Advances in neural information processing systems 36 (2023): 56027-56074.

**Questions:**

See Weaknesses.

---

> ### Author Response · Authors · 2025-11-21
>
> Thank you very much for your detailed review. We highly appreciate the extra effort you put into the analysis of our contribution and the related literature.
>
> We first want to mention that we have significantly reworked our experiment section, which was previously affected by poor design choices and a bug in the code. Instead of hand-picked PDEs, we now use general second-order PDEs with tunable parameters, covering the most used PDEs in specific parameter combinations. We also added a 2D real-world example, showing that our method can be readily applied in practice.
>
> With regards to your questions:
> 1. You mention that our results are known in the PDE community and that our method is essentially an application of the weighted conformal prediction algorithm of Barber et al. (2023). In both cases, you are absolutely correct. However, Barber’s algorithm is not a plug-and-play method, as it requires precise knowledge of the underlying distributions at every time step. This is exactly why it is rarely applied in time-series CP. We see our contribution in identifying the linear PDE class as a class where Barber’s method can actually be applied, and in providing a detailed analysis of when and how this is possible. While these results are known in the PDE community, the papers we cited that rely on the incorrect heuristic “small time step means small distribution shift” show that they are not widely known in the ML community.
>
> 2. You are also correct that our method does not (at least not directly) extend to non-linear PDEs. While we would welcome a broadly applicable CP method for PDEs, there is currently no approach that gives conformal guarantees even for linear PDEs—hence, we believe our work is an important starting point. All directly applicable approaches offer only asymptotic guarantees, where substantial undercoverage at individual time steps can still occur. This becomes relevant in settings like the heat equation, where a machine may fail once a critical temperature is reached, and only per-time-step guarantees can prevent such failures. To demonstrate that our method can be used in such scenarios, we added a real-world example in the appendix.
>
> Regarding your list of significant weaknesses:
>
> 1. **Description of figures and tables.**
> 	1. **Why 90% vs. 99%?**
> 	   This is stated in line 451 of the initial manuscript. The 90% refers to overall sample coverage, while the 99% refers to per-sample pointwise coverage. This convention was used in LSCI, and we kept it for a fair comparison. We agree it is confusing and led to initial overcoverage. In the revised version, we removed this distinction and consider a sample covered only if all of its points are covered.
> 	2. **Why does WCP’s line disappear?**
> 	   This is explained in the description of Figure 3 (lines 419–420). We stop WCP’s line once it predicts only infinite bands; otherwise it would show coverage of 1 while no longer producing meaningful intervals. In the new version, we use a dotted line showing the percentage of samples with infinite bandwidth.
> 	3. **What is in Table 1?**
> 	   This is described in lines 378–380 of the old manuscript. Could the reviewer clarify what exactly is causing the confusion so we can directly address it?
> 2. **Limited experiment setup.**
>    You are right that we initially only showed linear, univariate, synthetic experiments. In the updated version, we added a 2D real-world heat-equation dataset. We hope this demonstrates that our method is directly applicable in practice.
> 3. **Lack of comparison to baselines and literature discussion.**
>    This is a crucial point, and we appreciate the reviewer’s effort in highlighting it. The cited papers are relevant for UQ in PDEs, but none provide per–time-step guarantees (except [6], which requires bounds unavailable in practice). Papers [1,2] are non-conformal; papers [4,5] and most time-series approaches offer only asymptotic guarantees, meaning they may undercover at some time steps and overcover at others, with miscoverage vanishing only over an infinite horizon.
>    Our method instead provides theoretical guarantees at every time step. We deliberately refrain from comparing against these baselines because many require a different setup (e.g., recalibration at each step), and we do not claim to outperform them in terms of bandwidth or empirical coverage. We claim that our method is the only one achieving strict theoretical guarantees.
>    LSCI claim theoretical guarantees under local exchangeability; we show that this assumption typically does not hold. We added an extended discussion of this to the related literature section.
> 4. **WCP overcoverage.**
>    See "Why 90% vs. 99%?
>
> We acknowledge that, given the limited experiment section in the initial manuscript, the paper lacked practical value. We hope that the revised experiments, together with our discussion of current methods with conformal guarantees, address this concern and convince you to recommend the paper for publication.

---

### Official Review · Reviewer_fKy7 · 2025-11-01

**Soundness:** 4
**Presentation:** 4
**Contribution:** 3
**Rating:** 6
**Confidence:** 3

**Summary:**

This paper studies how conformal prediction (CP) fails under temporal non-stationarity common in time-dependent PDEs, where calibration and test data are not exchangeable. The authors first prove that, in function-space settings, even simple PDEs (e.g., the 1-D heat equation) produce mutually singular solution distributions across time, making exact coverage guarantees impossible. They then propose Weighted Conformal Prediction (WCP) for discretised PDE surrogates, deriving closed-form Gaussian densities for the evolving solutions and using likelihood-ratio weights to re-establish exact coverage. Experiments on fractional diffusion, backward heat, and reaction–diffusion equations show that WCP maintains nominal 90 % coverage over long horizons, whereas naïve CP and local-exchangeability (LSCI) methods quickly under-cover.

**Strengths:**

- Mathematical clarity. The analysis of mutual singularity in function spaces (Theorem 4.1) exposes a genuine limitation of applying CP directly to PDEs.

- Principled solution. The likelihood-weighted approach is a suitable approach to time-dependent distribution shift, avoiding unverifiable “local exchangeability” assumptions.

- Clear writing and sound experiments. The presentation is precise, proofs are rigorous, and empirical results clearly support the claims.

**Weaknesses:**

- Limited scope. Theoretical guarantees hold only for linear PDEs with Gaussian initial conditions, where analytic densities exist.

- Incremental novelty. Weighted CP under covariate shift is known (e.g., Barber et al., 2023); the contribution is mainly its application to PDE surrogates, not a new CP framework.

- Narrow experiments. Tests on 1-D synthetic PDEs demonstrate correctness but not scalability or real-world complexity.

**Questions:**

- How sensitive is WCP to deviations from Gaussianity or linearity?
- Can the framework be extended to nonlinear or stochastic PDEs without closed-form densities?

---

> ### Author Response · Authors · 2025-11-28
>
> We thank Reviewer fKy7 for the helpful feedback and the suggestions to improve the paper.
>
> We first want to mention that we have significantly reworked our experiment section, which previously was driven by poor design choices and a bug in the code. Instead of hand-picked specific PDEs, we switched to general second-order PDEs with tunable parameters, which cover the most used PDEs in specific parameter combinations.
> We also added a 2D real-world example, showing that our method can be readily applied in practice.
>
> Regarding your questions:
>
> 1. Deviations from Gaussianity stay minor as long as the distribution remains equivariant to linearity. We mentioned this briefly in the paper, but your comment motivated us to demonstrate it experimentally as well. We added experiments in the appendix using Laplace and logistic equations as starting points and show that our guarantees continue to hold. Outside the location–scale family, the method does not apply. Remark 4.3 explains why this limitation should rarely be an issue in practice.
> If the PDE itself deviates from linearity, the method is not applicable.
> 2. The framework does depend on closed-form densities. At this point, we do not see a straightforward way to extend it to non-linear PDEs or SDEs. This is beyond the scope of the paper, but we will keep the question in mind for future work.
>
> We also briefly address your broader concerns.
>
> 1. On the limited scope: In practice, we do not rely strictly on analytical expressions for the mean and variance, as they can be estimated reliably from the data (as already done in the paper). This makes the method usable even without an analytical description of the PDE.
> 2. Regarding the narrow experiments: We have greatly overworked our experiment section, which now includes a more rigorous analysis of the applicability to second-order PDEs, and we additionally added a 2D real-world example, which shows that our method can be readily applied in practice.
>
> We hope we could answer your questions. We further hope that we could address your concerns with regards to applicability in our answers and with the renewed experiment section. If that is the case, we would highly appreciate if you could strengthen your recommendation.

---

### Meta-Review · Area_Chair_s8sz · 2026-01-06

**Summary:**

The paper addresses the challenge of applying conformal prediction (CP) to time-dependent PDEs, where the exchangeability assumption is violated due to temporal distribution shifts. The authors prove that in function spaces, distributions at different times are mutually singular, rendering standard CP guarantees impossible. To mitigate this, they propose a Weighted Conformal Prediction (WCP) approach for discretized linear PDEs, leveraging the known Gaussian evolution of the solution to calculate exact density ratios for reweighting.

While the reviewers appreciated the mathematical clarity and the identification of the theoretical barrier in function spaces (Theorem 4.1), the consensus for rejection stems from three critical limitations:


1. Limited Scope and Applicability: The proposed method strictly relies on the linearity of the PDE and Gaussian (or location-scale) initial conditions to compute closed-form density ratios. Reviewers fKy7 and e72z noted that this excludes the vast majority of physically relevant, non-linear systems (e.g., Navier-Stokes) central to the Scientific ML community.

2. Incremental Novelty: The algorithmic contribution is primarily an application of the existing Weighted CP framework (Barber et al., 2023) to a specific class of problems where the distribution shift is analytically known. Reviewer e72z pointed out that the underlying theoretical results regarding PDE solution distributions are established in the PDE literature, limiting the methodological novelty for an ML conference.

3. Practical Utility: Reviewers (e.g., iHz7, e72z) raised significant concerns about the utility of the generated prediction bands. In cases of significant drift, the method maintains coverage by producing infinite (vacuous) bands. While this technically satisfies safety guarantees, it offers little informational value for practical engineering decision-making compared to existing asymptotic methods.

**Reviewer Concerns:**

Extension to Non-Linear Systems: The fundamental limitation remains that the method does not extend to non-linear PDEs without closed-form density transitions. The authors acknowledged this is beyond the scope, but for the reviewers, this severely limits the impact of the work.

Comparison to Baselines: Reviewer e72z noted a lack of comparison to standard time-series CP baselines. The authors argued these baselines only offer asymptotic guarantees, but the absence of empirical comparison makes it difficult to assess the trade-off between the proposed method's "exact but potentially vacuous" guarantees and the "approximate but informative" bands of existing methods.

Vacuous Bounds: Reviewer iHz7 remained unconvinced about the practical utility of a method that defaults to covering the entire space (infinite bandwidth) when the shift is large, questioning whether leveraging the specific linear structure provides a genuine advantage over standard techniques in difficult regimes.

**Reviewer Scores:**

Reviewer fKy7: 6 (Weak Accept) ->  6 (Weak Accept) . Reasoning: While initially positive about the clarity, the reviewer noted the "Limited scope" and "Incremental novelty." The rebuttal addressed experimental details but could not resolve the fundamental limitation to linear systems.

Reviewer e72z: 2 (Reject) -> 3 (Reject). Reasoning: The reviewer acknowledged the rebuttal efforts but maintained that the contribution was "not  novel" and the scope "limited." The fundamental critique regarding the lack of algorithmic contribution stands.

Reviewer iHz7: 2 (Reject) -> 3 (Reject). Reasoning: The reviewer explicitly stated their read became "more positive" after the rebuttal clarified the contribution regarding identifying the linear PDE class. However, they maintained that the results were "not especially compelling" due to the conservative/vacuous nature of the bands, preventing a full flip to acceptance.

---

### Decision · Program_Chairs · 2026-01-26

Reject